# Hierarchical Graph Representation Learning with Differentiable Pooling

**Rex Ying**
rexying@stanford.edu
Stanford University

**Jiaxuan You**
jiaxuan@stanford.edu
Stanford University

**Christopher Morris**
christopher.morris@udo.edu
TU Dortmund University

**Xiang Ren**
xiangren@usc.edu
University of Southern California

**William L. Hamilton**
wleif@stanford.edu
Stanford University

**Jure Leskovec**
jure@cs.stanford.edu
Stanford University

## Abstract

Recently, graph neural networks (GNNs) have revolutionized the field of graph representation learning through effectively learned node embeddings, and achieved state-of-the-art results in tasks such as node classification and link prediction. However, current GNN methods are inherently *flat* and do not learn *hierarchical* representations of graphs—a limitation that is especially problematic for the task of graph classification, where the goal is to predict the label associated with an entire graph. Here we propose DIFFPOOL, a differentiable graph pooling module that can generate hierarchical representations of graphs and can be combined with various graph neural network architectures in an end-to-end fashion. DIFFPOOL learns a differentiable soft cluster assignment for nodes at each layer of a deep GNN, mapping nodes to a set of clusters, which then form the coarsened input for the next GNN layer. Our experimental results show that combining existing GNN methods with DIFFPOOL yields an average improvement of 5–10% accuracy on graph classification benchmarks, compared to all existing pooling approaches, achieving a new state-of-the-art on four out of five benchmark data sets.

## 1   Introduction

In recent years there has been a surge of interest in developing graph neural networks (GNNs)—general deep learning architectures that can operate over graph structured data, such as social network data [16, 21, 36] or graph-based representations of molecules [7, 11, 15]. The general approach with GNNs is to view the underlying graph as a computation graph and learn neural network primitives that generate individual node embeddings by passing, transforming, and aggregating node feature information across the graph [15, 16]. The generated node embeddings can then be used as input to any differentiable prediction layer, e.g., for node classification [16] or link prediction [32], and the whole model can be trained in an end-to-end fashion.

However, a major limitation of current GNN architectures is that they are inherently *flat* as they only propagate information across the edges of the graph and are unable to infer and aggregate the information in a *hierarchical* way. For example, in order to successfully encode the graph structure of organic molecules, one would ideally want to encode the local molecular structure (e.g., individual

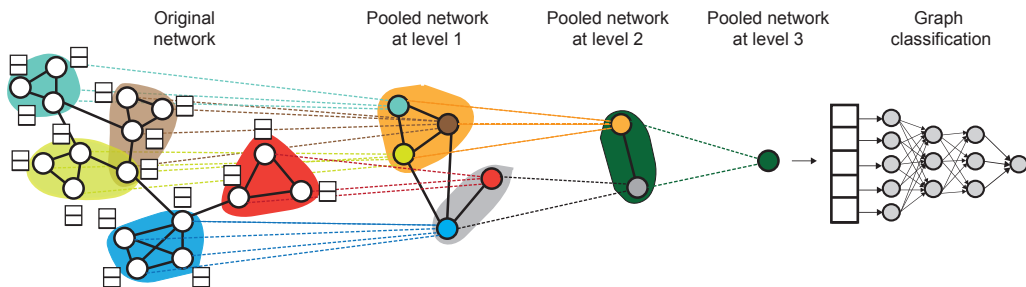

Figure 1: High-level illustration of our proposed method DIFFPOOL. At each hierarchical layer, we run a GNN model to obtain embeddings of nodes. We then use these learned embeddings to cluster nodes together and run another GNN layer on this coarsened graph. This whole process is repeated for $L$ layers and we use the final output representation to classify the graph.

atoms and their direct bonds) as well as the coarse-grained structure of the molecular graph (e.g., groups of atoms and bonds representing functional units in a molecule). This lack of hierarchical structure is especially problematic for the task of graph classification, where the goal is to predict the label associated with an *entire graph*. When applying GNNs to graph classification, the standard approach is to generate embeddings for all the nodes in the graph and then to *globally pool* all these node embeddings together, e.g., using a simple summation or neural network that operates over sets [7, 11, 15, 25]. This global pooling approach ignores any hierarchical structure that might be present in the graph, and it prevents researchers from building effective GNN models for predictive tasks over entire graphs.

Here we propose DIFFPOOL, a differentiable graph pooling module that can be adapted to various graph neural network architectures in an hierarchical and end-to-end fashion (Figure 1). DIFFPOOL allows for developing deeper GNN models that can learn to operate on hierarchical representations of a graph. We develop a graph analogue of the spatial pooling operation in CNNs [23], which allows for deep CNN architectures to iteratively operate on coarser and coarser representations of an image. The challenge in the GNN setting—compared to standard CNNs—is that graphs contain no natural notion of spatial locality, i.e., one cannot simply pool together all nodes in a "$m \times m$ patch" on a graph, because the complex topological structure of graphs precludes any straightforward, deterministic definition of a "patch". Moreover, unlike image data, graph data sets often contain graphs with varying numbers of nodes and edges, which makes defining a general graph pooling operator even more challenging.

In order to solve the above challenges, we require a model that learns how to cluster together nodes to build a hierarchical multi-layer scaffold on top of the underlying graph. Our approach DIFFPOOL learns a differentiable soft assignment at each layer of a deep GNN, mapping nodes to a set of clusters based on their learned embeddings. In this framework, we generate deep GNNs by "stacking" GNN layers in a hierarchical fashion (Figure 1): the input nodes at the layer $l$ GNN module correspond to the clusters learned at the layer $l - 1$ GNN module. Thus, each layer of DIFFPOOL coarsens the input graph more and more, and DIFFPOOL is able to generate a hierarchical representation of any input graph after training. We show that DIFFPOOL can be combined with various GNN approaches, resulting in an average 7% gain in accuracy and a new state of the art on four out of five benchmark graph classification tasks. Finally, we show that DIFFPOOL can learn interpretable hierarchical clusters that correspond to well-defined communities in the input graphs.

## 2   Related Work

Our work builds upon a rich line of recent research on graph neural networks and graph classification.

**General graph neural networks**. A wide variety of graph neural network (GNN) models have been proposed in recent years, including methods inspired by convolutional neural networks [5, 8, 11, 16, 21, 24, 29, 36], recurrent neural networks [25], recursive neural networks [1, 30] and loopy belief propagation [7]. Most of these approaches fit within the framework of "neural message passing" proposed by Gilmer *et al.* [15]. In the message passing framework, a GNN is viewed as a

message passing algorithm where node representations are iteratively computed from the features of their neighbor nodes using a differentiable aggregation function. Hamilton *et al.* [17] provides a conceptual review of recent advancements in this area, and Bronstein *et al.* [4] outlines connections to spectral graph convolutions.

**Graph classification with graph neural networks**. GNNs have been applied to a wide variety of tasks, including node classification [16, 21], link prediction [31], graph classification [7, 11, 40], and chemoinformatics [28, 27, 14, 19, 32]. In the context of graph classification—the task that we study here—a major challenge in applying GNNs is going from node embeddings, which are the output of GNNs, to a representation of the entire graph. Common approaches to this problem include simply summing up or averaging all the node embeddings in a final layer [11], introducing a "virtual node" that is connected to all the nodes in the graph [25], or aggregating the node embeddings using a deep learning architecture that operates over sets [15]. However, all of these methods have the limitation that they do not learn hierarchical representations (i.e., all the node embeddings are globally pooled together in a single layer), and thus are unable to capture the natural structures of many real-world graphs. Some recent approaches have also proposed applying CNN architectures to the concatenation of all the node embeddings [29, 40], but this requires a specifying (or learning) a canonical ordering over nodes, which is in general very difficult and equivalent to solving graph isomorphism.

Lastly, there are some recent works that learn hierarchical graph representations by combining GNNs with deterministic graph clustering algorithms [8, 35, 13], following a two-stage approach. However, unlike these previous approaches, we seek to *learn* the hierarchical structure in an end-to-end fashion, rather than relying on a deterministic graph clustering subroutine.

# 3 Proposed Method

The key idea of DIFFPOOL is that it enables the construction of deep, multi-layer GNN models by providing a differentiable module to hierarchically pool graph nodes. In this section, we outline the DIFFPOOL module and show how it is applied in a deep GNN architecture.

## 3.1 Preliminaries

We represent a graph $G$ as $(A, F)$, where $A \in \{0,1\}^{n \times n}$ is the adjacency matrix, and $F \in \mathbb{R}^{n \times d}$ is the node feature matrix assuming each node has $d$ features.[1] Given a set of labeled graphs $\mathcal{D} = \{(G_1, y_1), (G_2, y_2), ...\}$ where $y_i \in \mathcal{Y}$ is the label corresponding to graph $G_i \in \mathcal{G}$, the goal of graph classification is to learn a mapping $f : \mathcal{G} \to \mathcal{Y}$ that maps graphs to the set of labels. The challenge—compared to standard supervised machine learning setup—is that we need a way to extract useful feature vectors from these input graphs. That is, in order to apply standard machine learning methods for classification, e.g., neural networks, we need a procedure to convert each graph to an finite dimensional vector in $\mathbb{R}^D$.

**Graph neural networks**. In this work, we build upon graph neural networks in order to learn useful representations for graph classification in an end-to-end fashion. In particular, we consider GNNs that employ the following general "message-passing" architecture:

$$H^{(k)} = M(A, H^{(k-1)}; \theta^{(k)}), \tag{1}$$

where $H^{(k)} \in \mathbb{R}^{n \times d}$ are the node embeddings (i.e., "messages") computed after $k$ steps of the GNN and $M$ is the message propagation function, which depends on the adjacency matrix, trainable parameters $\theta^{(k)}$, and the node embeddings $H^{(k-1)}$ generated from the previous message-passing step.[2] The input node embeddings $H^{(0)}$ at the initial message-passing iteration $(k = 1)$, are initialized using the node features on the graph, $H^{(0)} = F$.

There are many possible implementations of the propagation function $M$ [15, 16]. For example, one popular variant of GNNs—Kipf's *et al.* [21] Graph Convolutional Networks (GCNs)—implements $M$ using a combination of linear transformations and ReLU non-linearities:

$$H^{(k)} = M(A, H^{(k-1)}; W^{(k)}) = \text{ReLU}(\tilde{D}^{-\frac{1}{2}} \tilde{A} \tilde{D}^{-\frac{1}{2}} H^{(k-1)} W^{(k-1)}), \tag{2}$$

where $\tilde{A} = A + I$, $\tilde{D} = \sum_j \tilde{A}_{ij}$ and $W^{(k)} \in \mathbb{R}^{d \times d}$ is a trainable weight matrix. The differentiable pooling model we propose can be applied to any GNN model implementing Equation (1), and is agnostic with regards to the specifics of how $M$ is implemented.

A full GNN module will run $K$ iterations of Equation (1) to generate the final output node embeddings $Z = H^{(K)} \in \mathbb{R}^{n \times d}$, where $K$ is usually in the range 2-6. For simplicity, in the following sections we will abstract away the internal structure of the GNNs and use $Z = \text{GNN}(A, X)$ to denote an arbitrary GNN module implementing $K$ iterations of message passing according to some adjacency matrix $A$ and initial input node features $X$.

**Stacking GNNs and pooling layers**. GNNs implementing Equation (1) are inherently flat, as they only propagate information across edges of a graph. The goal of this work is to define a general, end-to-end differentiable strategy that allows one to *stack* multiple GNN modules in a hierarchical fashion. Formally, given $Z = \text{GNN}(A, X)$, the output of a GNN module, and a graph adjacency matrix $A \in \mathbb{R}^{n \times n}$, we seek to define a strategy to output a new coarsened graph containing $m < n$ nodes, with weighted adjacency matrix $A^{'} \in \mathbb{R}^{m \times m}$ and node embeddings $Z^{'} \in \mathbb{R}^{m \times d}$. This new coarsened graph can then be used as input to another GNN layer, and this whole process can be repeated $L$ times, generating a model with $L$ GNN layers that operate on a series of coarser and coarser versions of the input graph (Figure 1). Thus, our goal is to learn how to cluster or pool together nodes using the output of a GNN, so that we can use this coarsened graph as input to another GNN layer. What makes designing such a pooling layer for GNNs especially challenging—compared to the usual graph coarsening task—is that our goal is not to simply cluster the nodes in one graph, but to provide a general recipe to hierarchically pool nodes across a broad set of input graphs. That is, we need our model to learn a pooling strategy that will generalize across graphs with different nodes, edges, and that can adapt to the various graph structures during inference.

## 3.2 Differentiable Pooling via Learned Assignments

Our proposed approach, DIFFPOOL, addresses the above challenges by learning a cluster assignment matrix over the nodes using the output of a GNN model. The key intuition is that we stack $L$ GNN modules and learn to assign nodes to clusters at layer $l$ in an end-to-end fashion, using embeddings generated from a GNN at layer $l - 1$. Thus, we are using GNNs to both extract node embeddings that are useful for graph classification, as well to extract node embeddings that are useful for hierarchical pooling. Using this construction, the GNNs in DIFFPOOL learn to encode a general pooling strategy that is useful for a large set of training graphs. We first describe how the DIFFPOOL module pools nodes at each layer given an assignment matrix; following this, we discuss how we generate the assignment matrix using a GNN architecture.

**Pooling with an assignment matrix**. We denote the learned cluster assignment matrix at layer $l$ as $S^{(l)} \in \mathbb{R}^{n_l \times n_{l+1}}$. Each row of $S^{(l)}$ corresponds to one of the $n_l$ nodes (or clusters) at layer $l$, and each column of $S^{(l)}$ corresponds to one of the $n_{l+1}$ clusters at the next layer $l + 1$. Intuitively, $S^{(l)}$ provides a soft assignment of each node at layer $l$ to a cluster in the next coarsened layer $l + 1$.

Suppose that $S^{(l)}$ has already been computed, i.e., that we have computed the assignment matrix at the $l$-th layer of our model. We denote the input adjacency matrix at this layer as $A^{(l)}$ and denote the input node embedding matrix at this layer as $Z^{(l)}$. Given these inputs, the DIFFPOOL layer $(A^{(l+1)}, X^{(l+1)}) = \text{DIFFPOOL}(A^{(l)}, Z^{(l)})$ coarsens the input graph, generating a new coarsened adjacency matrix $A^{(l+1)}$ and a new matrix of embeddings $X^{(l+1)}$ for each of the nodes/clusters in this coarsened graph. In particular, we apply the two following equations:

$$X^{(l+1)} = S^{(l)^T} Z^{(l)} \in \mathbb{R}^{n_{l+1} \times d}, \tag{3}$$

$$A^{(l+1)} = S^{(l)^T} A^{(l)} S^{(l)} \in \mathbb{R}^{n_{l+1} \times n_{l+1}}. \tag{4}$$

Equation (3) takes the node embeddings $Z^{(l)}$ and aggregates these embeddings according to the cluster assignments $S^{(l)}$, generating embeddings for each of the $n_{l+1}$ clusters. Similarly, Equation (4) takes the adjacency matrix $A^{(l)}$ and generates a coarsened adjacency matrix denoting the connectivity strength between each pair of clusters.

Through Equations (3) and (4), the DIFFPOOL layer coarsens the graph: the next layer adjacency matrix $A^{(l+1)}$ represents a coarsened graph with $n_{l+1}$ nodes or *cluster nodes*, where each individual

cluster node in the new coarsened graph corresponds to a cluster of nodes in the graph at layer $l$. Note that $A^{(l+1)}$ is a real matrix and represents a fully connected edge-weighted graph; each entry $A_{ij}^{(l+1)}$ can be viewed as the connectivity strength between cluster $i$ and cluster $j$. Similarly, the $i$-th row of $X^{(l+1)}$ corresponds to the embedding of cluster $i$. Together, the coarsened adjacency matrix $A^{(l+1)}$ and cluster embeddings $X^{(l+1)}$ can be used as input to another GNN layer, a process which we describe in detail below.

**Learning the assignment matrix**. In the following we describe the architecture of DIFFPOOL, i.e., how DIFFPOOL generates the assignment matrix $S^{(l)}$ and embedding matrices $Z^{(l)}$ that are used in Equations (3) and (4). We generate these two matrices using two separate GNNs that are both applied to the input cluster node features $X^{(l)}$ and coarsened adjacency matrix $A^{(l)}$. The *embedding GNN* at layer $l$ is a standard GNN module applied to these inputs:

$$Z^{(l)} = \text{GNN}_{l,\text{embed}}(A^{(l)}, X^{(l)}), \tag{5}$$

i.e., we take the adjacency matrix between the cluster nodes at layer $l$ (from Equation 4) and the pooled features for the clusters (from Equation 3) and pass these matrices through a standard GNN to get new embeddings $Z^{(l)}$ for the cluster nodes. In contrast, the *pooling GNN* at layer $l$, uses the input cluster features $X^{(l)}$ and cluster adjacency matrix $A^{(l)}$ to generate an assignment matrix:

$$S^{(l)} = \text{softmax}\left(\text{GNN}_{l,\text{pool}}(A^{(l)}, X^{(l)})\right), \tag{6}$$

where the softmax function is applied in a row-wise fashion. The output dimension of $\text{GNN}_{l,\text{pool}}$ corresponds to a pre-defined maximum number of clusters in layer $l$, and is a hyperparameter of the model.

Note that these two GNNs consume the same input data but have distinct parameterizations and play separate roles: The embedding GNN generates new embeddings for the input nodes at this layer, while the pooling GNN generates a probabilistic assignment of the input nodes to $n_{l+1}$ clusters.

In the base case, the inputs to Equations (5) and Equations (6) at layer $l = 0$ are simply the input adjacency matrix $A$ and the node features $F$ for the original graph. At the penultimate layer $L - 1$ of a deep GNN model using DIFFPOOL, we set the assignment matrix $S^{(L-1)}$ be a vector of 1's, i.e., all nodes at the final layer $L$ are assigned to a single cluster, generating a final embedding vector corresponding to the entire graph. This final output embedding can then be used as feature input to a differentiable classifier (e.g., a softmax layer), and the entire system can be trained end-to-end using stochastic gradient descent.

**Permutation invariance**. Note that in order to be useful for graph classification, the pooling layer should be invariant under node permutations. For DIFFPOOL we get the following positive result, which shows that any deep GNN model based on DIFFPOOL is permutation invariant, as long as the component GNNs are permutation invariant.

**Proposition 1.** *Let* $P \in \{0, 1\}^{n \times n}$ *be any permutation matrix, then* $\text{DIFFPOOL}(A, Z) = \text{DIFFPOOL}(PAP^T, PX)$ *as long as* $\text{GNN}(A, X) = \text{GNN}(PAP^T, X)$ *(i.e., as long as the GNN method used is permutation invariant).*

*Proof.* Equations (5) and (6) are permutation invariant by the assumption that the GNN module is permutation invariant. And since any permutation matrix is orthogonal, applying $P^T P = I$ to Equation (3) and (4) finishes the proof. $\qquad\square$

## 3.3 Auxiliary Link Prediction Objective and Entropy Regularization

In practice, it can be difficult to train the pooling GNN (Equation 4) using only gradient signal from the graph classification task. Intuitively, we have a non-convex optimization problem and it can be difficult to push the pooling GNN away from spurious local minima early in training. To alleviate this issue, we train the pooling GNN with an auxiliary link prediction objective, which encodes the intuition that nearby nodes should be pooled together. In particular, at each layer $l$, we minimize $L_{\text{LP}} = ||A^{(l)}, S^{(l)} S^{(l)^T}||_F$, where $|| \cdot ||_F$ denotes the Frobenius norm. Note that the adjacency matrix $A^{(l)}$ at deeper layers is a function of lower level assignment matrices, and changes during training.

Another important characteristic of the pooling GNN (Equation 4) is that the output cluster assignment for each node should generally be close to a one-hot vector, so that the membership for each cluster or subgraph is clearly defined. We therefore regularize the entropy of the cluster assignment by minimizing $L_{\text{E}} = \frac{1}{n} \sum_{i=1}^{n} H(S_i)$, where $H$ denotes the entropy function, and $S_i$ is the $i$-th row of $S$.

During training, $L_{\text{LP}}$ and $L_{\text{E}}$ from each layer are added to the classification loss. In practice we observe that training with the side objective takes longer to converge, but nevertheless achieves better performance and more interpretable cluster assignments.

## 4  Experiments

We evaluate the benefits of DIFFPOOL against a number of state-of-the-art graph classification approaches, with the goal of answering the following questions:

**Q1** How does DIFFPOOL compare to other pooling methods proposed for GNNs (e.g., using sort pooling [40] or the SET2SET method [15])?
**Q2** How does DIFFPOOL combined with GNNs compare to the state-of-the-art for graph classification task, including both GNNs and kernel-based methods?
**Q3** Does DIFFPOOL compute meaningful and interpretable clusters on the input graphs?

**Data sets**. To probe the ability of DIFFPOOL to learn complex hierarchical structures from graphs in different domains, we evaluate on a variety of relatively large graph data sets chosen from benchmarks commonly used in graph classification [20]. We use protein data sets including ENZYMES, PROTEINS [3, 12], D&D [10], the social network data set REDDIT-MULTI-12K [39], and the scientific collaboration data set COLLAB [39]. See Appendix A for statistics and properties. For all these data sets, we perform 10-fold cross-validation to evaluate model performance, and report the accuracy averaged over 10 folds.

**Model configurations**. In our experiments, the GNN model used for DIFFPOOL is built on top of the GRAPHSAGE architecture, as we found this architecture to have superior performance compared to the standard GCN approach as introduced in [21]. We use the "mean" variant of GRAPHSAGE [16] and apply a DIFFPOOL layer after every two GRAPHSAGE layers in our architecture. A total of 2 DIFFPOOL layers are used for the datasets. For small datasets such as ENZYMES, PROTEINS and COLLAB, 1 DIFFPOOL layer can achieve similar performance. After each DIFFPOOL layer, 3 layers of graph convolutions are performed, before the next DIFFPOOL layer, or the readout layer. The embedding matrix and the assignment matrix are computed by two separate GRAPHSAGE models respectively. In the 2 DIFFPOOL layer architecture, the number of clusters is set as 25% of the number of nodes before applying DIFFPOOL, while in the 1 DIFFPOOL layer architecture, the number of clusters is set as 10%. Batch normalization [18] is applied after every layer of GRAPHSAGE. We also found that adding an $\ell_2$ normalization to the node embeddings at each layer made the training more stable. In Section 4.2, we also test an analogous variant of DIFFPOOL on the STRUCTURE2VEC [7] architecture, in order to demonstrate how DIFFPOOL can be applied on top of other GNN models. All models are trained for 3 000 epochs with early stopping applied when the validation loss starts to drop. We also evaluate two simplified versions of DIFFPOOL:
- DIFFPOOL-DET, is a DIFFPOOL model where assignment matrices are generated using a deterministic graph clustering algorithm [9].
- DIFFPOOL-NOLP is a variant of DIFFPOOL where the link prediction side objective is turned off.

### 4.1  Baseline Methods

In the performance comparison on graph classification, we consider baselines based upon GNNs (combined with different pooling methods) as well as state-of-the-art kernel-based approaches.

**GNN-based methods**.
- GRAPHSAGE with global mean-pooling [16]. Other GNN variants such as those proposed in [21] are omitted as empirically GraphSAGE obtained higher performance in the task.
- STRUCTURE2VEC (S2V) [7] is a state-of-the-art graph representation learning algorithm that combines a latent variable model with GNNs. It uses global mean pooling.
- Edge-conditioned filters in CNN for graphs (ECC) [35] incorporates edge information into the GCN model and performs pooling using a graph coarsening algorithm.

Table 1: Classification accuracies in percent. The far-right column gives the relative increase in accuracy compared to the baseline GRAPHSAGE approach.

| | Method | Data Set | | | | | |
|---|---|---|---|---|---|---|---|
| | | ENZYMES | D&D | REDDIT-MULTI-12K | COLLAB | PROTEINS | Gain |
| Kernel | GRAPHLET | 41.03 | 74.85 | 21.73 | 64.66 | 72.91 | |
| | SHORTEST-PATH | 42.32 | 78.86 | 36.93 | 59.10 | 76.43 | |
| | 1-WL | 53.43 | 74.02 | 39.03 | 78.61 | 73.76 | |
| | WL-OA | 60.13 | 79.04 | 44.38 | 80.74 | 75.26 | |
| GNN | PATCHYSAN | – | 76.27 | 41.32 | 72.60 | 75.00 | 4.17 |
| | GRAPHSAGE | 54.25 | 75.42 | 42.24 | 68.25 | 70.48 | – |
| | ECC | 53.50 | 74.10 | 41.73 | 67.79 | 72.65 | 0.11 |
| | SET2SET | 60.15 | 78.12 | 43.49 | 71.75 | 74.29 | 3.32 |
| | SORTPOOL | 57.12 | 79.37 | 41.82 | 73.76 | 75.54 | 3.39 |
| | DIFFPOOL-DET | 58.33 | 75.47 | 46.18 | **82.13** | 75.62 | 5.42 |
| | DIFFPOOL-NOLP | 61.95 | 79.98 | 46.65 | 75.58 | 76.22 | 5.95 |
| | DIFFPOOL | **62.53** | **80.64** | **47.08** | 75.48 | **76.25** | **6.27** |

- PATCHYSAN [29] defines a receptive field (neighborhood) for each node, and using a canonical node ordering, applies convolutions on linear sequences of node embeddings.
- SET2SET replaces the global mean-pooling in the traditional GNN architectures by the aggregation used in SET2SET [38]. Set2Set aggregation has been shown to perform better than mean pooling in previous work [15]. We use GRAPHSAGE as the base GNN model.
- SORTPOOL [40] applies a GNN architecture and then performs a single layer of soft pooling followed by 1D convolution on sorted node embeddings.

For all the GNN baselines, we use 10-fold cross validation numbers reported by the original authors when possible. For the GRAPHSAGE and SET2SET baselines, we use the base implementation and hyperparameter sweeps as in our DIFFPOOL approach. When baseline approaches did not have the necessary published numbers, we contacted the original authors and used their code (if available) to run the model, performing a hyperparameter search based on the original author's guidelines.

**Kernel-based algorithms**. We use the GRAPHLET [34], the SHORTEST-PATH [2], WEISFEILER-LEHMAN kernel (WL) [33], and WEISFEILER-LEHMAN OPTIMAL ASSIGNMENT KERNEL (WL-OA) [22] as kernel baselines. For each kernel, we computed the normalized gram matrix. We computed the classification accuracies using the $C$-SVM implementation of LIBSVM [6], using 10-fold cross validation. The $C$ parameter was selected from $\{10^{-3}, 10^{-2}, \ldots, 10^2, 10^3\}$ by 10-fold cross validation on the training folds. Moreover, for WL and WL-OA we additionally selected the number of iteration from $\{0, \ldots, 5\}$.

## 4.2 Results for Graph Classification

Table 1 compares the performance of DIFFPOOL to these state-of-the-art graph classification baselines. These results provide positive answers to our motivating questions **Q1** and **Q2**: We observe that our DIFFPOOL approach obtains the highest average performance among all pooling approaches for GNNs, improves upon the base GRAPHSAGE architecture by an average of $6.27\%$, and achieves state-of-the-art results on 4 out of 5 benchmarks. Interestingly, our simplified model variant, DIFFPOOL-DET, achieves state-of-the-art performance on the COLLAB benchmark. This is because many collaboration graphs in COLLAB show only single-layer community structures, which can be captured well with pre-computed graph clustering algorithm [9]. One observation is that despite significant performance improvement, DIFFPOOL could be unstable to train, and there is significant variation in accuracy across different runs, even with the same hyperparameter setting. It is observed that adding the link predictioin objective makes training more stable, and reduces the standard deviation of accuracy across different runs.

**Differentiable Pooling on STRUCTURE2VEC**. DIFFPOOL can be applied to other GNN architectures besides GRAPHSAGE to capture hierarchical structure in the graph data. To further support answering **Q1**, we also applied DIFFPOOL on Structure2Vec (S2V). We ran experiments using S2V with three layer architecture, as reported in [7]. In the first variant, one DIFFPOOL layer is applied after the first layer of S2V, and two more S2V layers are stacked on top of the output of DIFFPOOL.

The second variant applies one DIFFPOOL layer after the first and second layer of S2V respectively. In both variants, S2V model is used to compute the embedding matrix, while GRAPHSAGE model is used to compute the assignment matrix.

Table 2: Accuracy results of applying DIFFPOOL to S2V.

| Data Set | Method | | |
|---|---|---|---|
| | S2V | S2V WITH 1 DIFFPOOL | S2V WITH 2 DIFFPOOL |
| ENZYMES | 61.10 | 62.86 | **63.33** |
| D&D | 78.92 | 80.75 | **82.07** |

The results in terms of classification accuracy are summarized in Table 2. We observe that DIFFPOOL significantly improves the performance of S2V on both ENZYMES and D&D data sets. Similar performance trends are also observed on other data sets. The results demonstrate that DIFFPOOL is a general strategy to pool over hierarchical structure that can benefit different GNN architectures.

**Running time**. Although applying DIFFPOOL requires additional computation of an assignment matrix, we observed that DIFFPOOL did not incur substantial additional running time in practice. This is because each DIFFPOOL layer reduces the size of graphs by extracting a coarser representation of the graph, which speeds up the graph convolution operation in the next layer. Concretely, we found that GRAPHSAGE with DIFFPOOL was $12\times$ faster than the GRAPHSAGE model with SET2SET pooling, while still achieving significantly higher accuracy on all benchmarks.

### 4.3 Analysis of Cluster Assignment in DIFFPOOL

**Hierarchical cluster structure**. To address **Q3**, we investigated the extent to which DIFFPOOL learns meaningful node clusters by visualizing the cluster assignments in different layers. Figure 2 shows such a visualization of node assignments in the first and second layers on a graph from COLLAB data set, where node color indicates its cluster membership. Node cluster membership is determined by taking the argmax of its cluster assignment probabilities. We observe that even when learning cluster assignment based solely on the graph classification objective, DIFFPOOL can still capture the hierarchical community structure. We also observe significant improvement in membership assignment quality with link prediction auxiliary objectives.

**Dense vs. sparse subgraph structure**. In addition, we observe that DIFFPOOL learns to collapse nodes into soft clusters in a non-uniform way, with a tendency to collapse densely-connected subgraphs into clusters. Since GNNs can efficiently perform message-passing on dense, clique-like subgraphs (due to their small diameters) [26], pooling together nodes in such a dense subgraph is not likely to lead to any loss of structural information. This intuitively explains why collapsing dense subgraphs is a useful pooling strategy for DIFFPOOL. In contrast, sparse subgraphs may contain many interesting structures, including path-, cycle- and tree-like structures, and given the high-diameter induced by sparsity, GNN message-passing may fail to capture these structures. Thus, by separately pooling distinct parts of a sparse subgraph, DIFFPOOL can learn to capture the meaningful structures present in sparse graph regions (e.g., as in Figure 2).

**Assignment for nodes with similar representations**. Since the assignment network computes the soft cluster assignment based on features of input nodes and their neighbors, nodes with both similar input features and neighborhood structure will have similar cluster assignment. In fact, one can construct synthetic cases where 2 nodes, although far away, have exactly the same neighborhood structure and features for self and all neighbors. In this case the pooling network is forced to assign them into the same cluster, which is different from the concept of pooling in other architectures such as image ConvNets. In some cases we do observe that disconnected nodes are pooled together.

In practice we rely on the identifiability assumption similar to Theorem 1 in GraphSAGE [16], where nodes are identifiable via their features. This holds in many real datasets [3]. The auxiliary link prediction objective is observed to also help discouraging nodes that are far away to be pooled together. Furthermore, it is possible to use more sophisticated GNN aggregation function such as

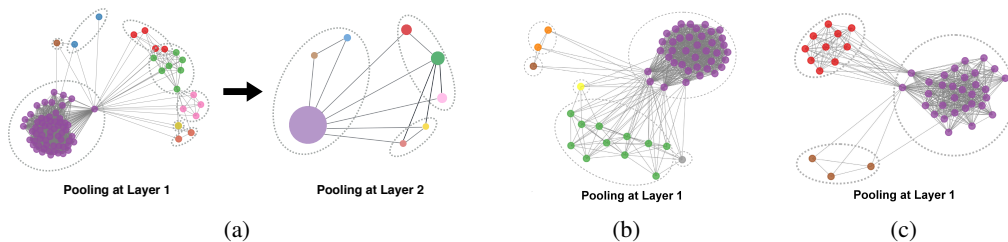

Figure 2: Visualization of hierarchical cluster assignment in DIFFPOOL, using example graphs from COLLAB. The left figure (a) shows hierarchical clustering over two layers, where nodes in the second layer correspond to clusters in the first layer. (Colors are used to connect the nodes/clusters across the layers, and dotted lines are used to indicate clusters.) The right two plots (b and c) show two more examples first-layer clusters in different graphs. Note that although we globally set the number of clusters to be 25% of the nodes, the assignment GNN automatically learns the appropriate number of meaningful clusters to assign for these different graphs.

high-order moments [37] to distinguish nodes that are similar in structure and feature space. The overall framework remains unchanged.

**Sensitivity of the Pre-defined Maximum Number of Clusters**. We found that the assignment varies according to the depth of the network and $C$, the maximum number of clusters. With larger $C$, the pooling GNN can model more complex hierarchical structure. The trade-off is that very large $C$ results in more noise and less efficiency. Although the value of $C$ is a pre-defined parameter, the pooling net learns to use the appropriate number of clusters by end-to-end training. In particular, some clusters might not be used by the assignment matrix. Column corresponding to unused cluster has low values for all nodes. This is observed in Figure 2(c), where nodes are assigned predominantly into 3 clusters.

## 5 Conclusion

We introduced a differentiable pooling method for GNNs that is able to extract the complex hierarchical structure of real-world graphs. By using the proposed pooling layer in conjunction with existing GNN models, we achieved new state-of-the-art results on several graph classification benchmarks. Interesting future directions include learning hard cluster assignments to further reduce computational cost in higher layers while also ensuring differentiability, and applying the hierarchical pooling method to other downstream tasks that require modeling of the entire graph structure.

## Acknowledgement

This research has been supported in part by DARPA SIMPLEX, Stanford Data Science Initiative, Huawei, JD and Chan Zuckerberg Biohub. Christopher Morris is funded by the German Science Foundation (DFG) within the Collaborative Research Center SFB 876 "Providing Information by Resource-Constrained Data Analysis", project A6 "Resource-efficient Graph Mining". The authors also thank Marinka Zitnik for help in visualizing the high-level illustration of the proposed methods.

## Footnotes

[1] We do not consider edge features, although one can easily extend the algorithm to support edge features using techniques introduced in [35].

[2] For notational convenience, we assume that the embedding dimension is $d$ for all $H^{(k)}$; however, in general this restriction is not necessary.

[3]However, some chemistry molecular graph datasets contain many nodes that are structurally similar, and assignment network is observed to pool together nodes that are far away.

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
