[Reviews · NeurIPS 2018]

Reviewer 1



the paper introduces a differentiable clustering operation enabling neural methods acting on graphs to process information in a hierarchical fashion in analogy to CNN pooling layers (but preserving graph symmetry properties). The paper is very well written, the method is simple (which is not a bad thing!) and widely applicable, and the results are favourable. Building hierarchical representations is an important feature of CNNs and extending this to more general graph datatypes is clearly an important contribution. Overall I strongly support publication of this paper. I only have minor additional thoughts: 1. When the edges have labels (e.g. single/double/triple bonds in the chemistry example used in the motivation of the method), would equation (4) be extended to transform adjacency matrices for each edge type separately by S? In this case the link prediction objective will need altering. Maybe including an experiment with labelled edges would make the presentation of this otherwise simple method more complete? 2. You have to manually set the maximum number of clusters in each layer (please make this clearer near equation 6 - currently the only explicit reference to this fact is at the end of the caption of Fig2). It is nice that the GNN can learn to use less than the maximum number of clusters, but how sensitive are the results to the cluster count hyperparameters? Are the examples in Fig2 compelling because there is obvious clustering in the graphs? What does the clustering figure look like for data where the clusters are not so obvious?

Reviewer 2



In this paper, the task of learning the hierarchical representation of graphs is achieved by stacking GNNs and Pooling layers. The authors first specify the difficulty in stacking GNNs and Pooling layers then propose a differentiable pooling approach called DIFFPOOL to deal with this problem. The paper is well-written and easily to follow. Besides providing the theoretical formalization and analysis, the authors conducting sufficient experiments to proved that the proposed method outperformed state-of-the-art baselines in five popular tasks. While the number of classes increases to 11 in the dataset named by REDDIT-MULTI-12K, the accuracy (in percent) is quite low, at under 50%. The effects of DIFFPOOL are visualized clearly in Figure 2. A wide range of references are given. Although the intuition of using hierarchical representation in classification was appeared in previous research such as “Convolutional neural networks on graphs with fast localized spectral filtering” [8], it is worth noting that the proposed model in this paper are able to learn the hierarchical representation end-to-end. Overall, the contribution to the literature of neural graph nets is clear and significant.

Reviewer 3



This paper presents a hierarchical graph representation learning algorithm with the novel differentiable pooling layers. Such differentiable pooling layers learn node assignments through standard GNNs, and then based on such node assignments pool and update node representations using such assignments. Experiments on a few benchmark graph classification tasks show consistent improvement over non-hierarchical baselines. The differentiable pooling idea proposed in this paper is a nice contribution, and I can see ideas based on this be very useful for learning hierarchical and more and more abstract representations for graphs by pooling on them. The empirical results presented in the paper also seems convincing. One question I had about the proposed approach is that, the current approach would assign nodes with similar representations to similar higher level nodes. This is quite different from the pooling in convnets, which groups pixels that are nearby, rather than grouping pixels that are similar. Do you ever see non-local nodes be grouped together in DiffPool? Is this type of pooling by similarity better or worse than pooling by locality? Overall I quite liked this paper, the presentation is clear and the contribution is sound. I would recommend to accept this paper.